# A Comparative Analysis on the Effect of Variety of Grape Pomace Extracts on the Ice-Templated 3D Cryogel Features

**DOI:** 10.3390/gels7030076

**Published:** 2021-06-23

**Authors:** Irina Elena Raschip, Nicusor Fifere, Maria Valentina Dinu

**Affiliations:** Petru Poni Institute of Macromolecular Chemistry, Grigore Ghica Voda Alley 41A, 700487 Iasi, Romania; fifere.nicusor@icmpp.ro (N.F.); vdinu@icmpp.ro (M.V.D.)

**Keywords:** grape pomace extract, Feteasca Neagra and Merlot grape variety, 3D xanthan-based composites, surface properties, radical scavenging activity, microbiological features

## Abstract

Nowadays, there is a growing interest in developing functional packaging materials from renewable resources containing bioactive compounds (such as polyphenols) in order to reduce the use of petroleum-based plastics and their impact on the environment. In this regard, the effect of a variety and concentration of grape pomace extracts (Feteasca Neagra or Merlot) incorporated within ice-templated 3D xanthan-based composites was evaluated by considering their water content, surface and texture properties, radical scavenging and microbiological activities. The embedding of Feteasca Neagra or Merlot grape pomace extracts was studied by static water swelling and contact angle measurements, and SEM, EDX, and TGA analyses. The water contact angle results showed an increase in the surface hydrophobicity of the extract-loaded cryogels with an increase in extract content from 10 to 40 *v*/*v*%. SEM micrographs indicated that the entrapment of grape pomace extracts affected the morphology of the pore walls and reduced the pore sizes. The antioxidant activity of grape pomace extract-loaded composite cryogels was closely related to the total phenolic content of grape variety and to their concentration into matrices. The highly hydrophobic character of composite cryogels containing Merlot grape pomace extract and their remarkable antimicrobial activity indicates a great potential of these materials for food packaging applications.

## 1. Introduction

Pomace is the main by-product of wine and juice production, representing the residue left after the juice is extracted by pressing the grapes during the vinification process. It is produced in huge quantities in many parts of the world and is composed mainly of skins, seeds and sometimes bunches.

Currently, the pomace from the winemaking is used as feed, fertilizer or can be discarded (but this involves a cost paid by the winemaker) [1,2].

Pomace is rich in bioactive compounds, such as polyphenols, that have antioxidant and antimicrobial properties. Therefore, winery residues can be an alternative source of natural antioxidants. Garrido et al. [3] showed that pomace extracts obtained from red grapes added to pork burgers, prevent lipid oxidation, and increase color stability and product acceptability. Tseng and Zhao [4] demonstrated that the wine resulting from the vinification process is an alternative source of antioxidant compounds and dietary fiber, in addition to decreasing the degree of oxidation of lipids in samples during the refrigeration of yogurts and sauces (dressings) for salads.

By-products from winemaking, such as pomace, are sources of antioxidant compounds and have not been sufficiently exploited so far but are of increasing industrial interest [2]. Because it is often stored in open areas, leding to serious environmental problems (for instance, phytotoxicity, toxicity to aquatic organisms and suppression of soil micro-organisms), its exploitation as a source of antioxidants in obtaining and/or improving eco-friendly packaging is an environmental benefit. 

The packaging industry continues to be one of the important and innovative areas in the development of production processes because, in addition to the need for food protection, the physicochemical requirements and the attractive aesthetic functions of the preservation (storage) system are also important. Nowadays, these aims are mainly fulfilled through vid treatments use, the sterilization of materials and the production of biological packaging with specific functionalities, such as antimicrobial and antioxidant properties. 

The use of compounds with antioxidant and antibacterial activity in the food industry has become increasingly necessary to ensure high food quality. In this regard, a plethora of research studies are concentrated on finding new alternatives for synthetic-based plastic films. A very promising one is that based on polysaccharides as raw biodegradable materials to improve food preservation and reduce the use of chemical additives [5,6,7].

The good film-forming capacity of several polysaccharides (chitosan, alginate, xanthan, dextran, starch, curdlan, and pullulan) could lead to new opportunities to expand the field of food packaging systems [7,8,9]. For example, black chokeberry (*Aronia melanocarpa*) pomace extracts [10], tannic acid [11], rutin [12], apple peel polyphenols [13], and grape seed extract [14] have been incorporated within chitosan-based systems. Hybrid composite films consisting of chitosan, xylan, polyvinyl alcohol (PVA), hydroxyapatite, and curcumin, as antioxidant agent, with improved antimicrobial and antifungal against *A. niger* and *A. flavus* have been also reported [15].

Recently, Golly et al. [16] focused on the shelf-life extension of Pinot noir grapes under cold temperature storage, using xanthan (XG)-based coatings enriched with ascorbic and citric acids. It was shown that the citric acid-enriched XG-based coatings were the best in preserving the texture, color, and antioxidant properties of Pinot noir grapes under cold temperature storage. In the research done by Sharma and Ramana Rao [17], XG-based coatings enriched with cinnamic acid prevented browning and extended shelf-life up to 4 days and 8 days of fresh-cut “Nashpati” and “Babugosha” pears by inhibiting enzyme activity and thus delaying the browning incidence. XG microparticles loaded with *Eschweilera nana Miersleaves* extract obtained by the spray-drying technique developed a remarkably improved antioxidant capacity [18]. Polymeric nanoparticles containing α-tocopherol and incorporated into a XG-based matrix had a great potential for preserving the physicochemical characteristics of fresh-cut red delicious apples during 21 days of storage [19].

Consequently, the development of sustainable composites has been a priority for polymer engineering research over the past two decades [20,21]. The composites are made of plastics based on organic components in combination with new natural fillers. Natural fillers act as hardeners in the matrix material to improve the mechanical properties of the mixtures. Many types of natural fillers have been studied for these purposes, such as perennial herbs [21], bamboo [22], quercetin, anthocyanins, tannin, and lipoic acid [23,24]. However, so far, limited attention has been paid to residues generated by the food processing industry, such as apples, tomatoes and grape pomace. Nevertheless, there is some research on the use of fatty acids in tomatoes to create polymers [25,26]. Based on the chemical properties of these materials and the constant character of these by-products, pomace from various fruits and vegetables offer substantial potential as a cellulosic material used to obtain various composites [20]. 

Recently, in our group, physically cross-linked XG/PVA cryogels embedding freeze-dried red grape pomace as powders were designed [27,28]. The optimum preparation conditions were found by systematic modification of the ratio between XG and PVA, the number and the temperature of the freeze/thawing cycles. Our previous study demonstrated the benefits of using pomace to improve the antioxidant and antimicrobial properties of XG/PVA films [27]. However, the effect of the variety of grape pomace extracts on the water swelling, surface and texture properties, radical scavenging and microbiological activities of the XG/PVA-based cryogels has not been studied. Consequently, in this paper, we maintained constant the synthesis conditions and the ratio between polymeric components, and we studied the effect of the grape variety (Feteasca Neagra or Merlot) and their concentrations on the ice-templated 3D cryogel features. This investigation is relevant because the composition, antioxidant and antibacterial activities of the grape pomace extracts depend on (i) the grape variety; (ii) the vinification process; (iii) the location of the vine; and (iv) the climatic conditions. In addition, the use of grape pomace (known as residues in food industry activities) to obtain new composite materials offers a green alternative to fillers or synthetic fibers and can contribute to the development of a sustainable “circular economy”.

## 2. Results and Discussion

### 2.1. Preparation and Characterization of Ice-Templated 3D Cryogels

In our study, the polymer matrix was composed of XG with a relative viscosity-average molecular weight of 1.98 × 10^6^ g mol^−1^ and polyvinyl alcohol (PVA) with a number-average molar mass of 96,619 g mol^−1^ and polydispersity index of 1.17. The grape pomace extracts derived from Feteasca Neagra or Merlot grape type were incorporated into the XG/PVA matrix in order to establish which variety of grapes endows the polymeric films with special properties. The 3D XG-based composite cryogels were prepared by the freeze/thawing approach, using the optimum reaction conditions identified previously (1:1 weight ratio between XG and PVA; and seven cycles of freeze/thawing) [27]. In this study, the grape variety (Feteasca Neagra or Merlot) and their content within the XG/PVA networks (10 to 40 *v*/*v*%) were the only synthesis parameters modified. The sample codes, their composition, and some characteristics are listed in Table 1. Each sample code includes “CG” from the composite gel and “F” or “M” from the variety of the grape pomace extract, Feteasca Neagra or Merlot; the final number from the sample codes represents the amount of extract entrapped within the composite matrices. Optical images of the samples dried either by lyophilization for 24 h, at −50 °C and 0.05 mbars (images A and B) or by solvent casting at 40 °C in an oven for 24 h (images C and D) are presented in Figure 1.

As Table 1 shows, the presence of the grape pomace extracts within the polymeric matrix was proved by the significant decrease in the water absorbency, i.e., SR values. The composite films containing grape pomace extracts were stable and did not break apart during the swelling tests, in contrast to the pure XG cryogel film or the XG/PVA composite without bioactive compounds, which fell into pieces after some time. 

The composite samples containing the highest amount of grape pomace extract (samples CG.F30 for Feateasca Neagra grape type and CG.M40 for Merlot grape variety, Table 1) exhibited the least swelling: 15.8 g/g and, respectively, 11.19 g/g, which means almost 2 and 3 times less water uptake, compared to the neat XG/PVA composite film without the extract (sample CG, Table 1). The decrease of the SR values with the addition of grape pomace extract within XG/PVA networks indicates the existence of intermolecular interactions between the functional groups of the polymeric components and those from extracts, which led to fewer hydrophilic sites for H_2_O molecules. A similar swelling behavior was previously reported for chitosan-based films containing extracts from black chokeberry pomace [10].

The hydrophobic and hydrophilic character of the surface of the XG/PVA-based cryogels without or with extract was further evaluated by performing water contact angle (θ, degrees) measurements (Figure 2).

θ > 65° correspond to a hydrophobic surface, while θ < 65° mean a hydrophilic surface [29]. Generally, for food packaging applications, highly hydrophobic films are desirable since they maintain the quality and prolong the shelf-life of products [12]. The water contact angle (θ, degrees) results, presented in Figure 2, showed an increased hydrophobicity of the extract-loaded cryogel films with the increase of the extract content from 10 to 40 *v*/*v*%. The significant improvement of surface hydrophobicity for the cryogel films was assigned to the incorporation of polyphenols, i.e., *trans*-resveratrol, the major compound determined by HPLC (see Section 4, Materials and Methods), which forms hydrogen bonding with the XG/PVA matrix. Similar results were reported for the entrapping of mango leaf extract [30] or rutin extract [12] within chitosan films. 

The presence of embedded grape pomace extracts within the XG/PVA composite matrix was also proved by EDX analysis of the composites (Figure 3 and Figure 4).

A semi-quantitative elemental analysis on the surface of CG.F10, CG.F.20, CG.F30 (Figure 3), CG.M10, CG.M20, CG.M30, and CG.M40 (Figure 4) showed the presence of Na, Mg, P, K, Zn from grape pomace extracts and C and O from polymeric compounds. As expected, the content (wt%) of minerals increased on the surface of films with the increase in the amount of extract incorporated within the XG/PVA networks. 

The existence of minerals within the samples containing grape pomace extracts was also supported by the TGA analysis. Thus, to check this, two cryogel samples (CG and CG.F10) were examined by TGA (Appendix A and Table 2). 

The study of XG/PVA composite cryogels embedding different concentrations of pomace extracts varieties (Feteasca Neagra or Merlot) and the establishment of antioxidant and antibacterial activities are essential to determine their potential field of applications. Therefore, in the next section, the antioxidant and antimicrobial properties are described.

### 2.2. Effect of Grape Pomace Extracts Type and Their Amount on the Biological Activity of Composites

#### 2.2.1. Antioxidant Activity

The 2,2-diphenyl-1-picrylhydrazyl (DPPH) radical scavenging activity of cryogel composites entrapping Feteasca Neagra or Merlot grape pomace extracts is shown in Figure 5 and Figure 6. 

The film samples with Feteasca Neagra and Merlot grape pomace extracts showed significant differences in their antioxidant activity, which were concentration dependent. The control films (without extracts) did not show antioxidant activity. The films with Feteasca Neagra type grape pomace extract (CG.F10–CG.F30) proved to be the most active, compared to the samples containing Merlot type grape extract (CG.M10–CG.M40) (Figure 5 and Figure 6). This result is explained by the higher phenolic content of Feteasca Neagra extract in contrast to that of Merlot extract [31]. 

The films with Feteasca Neagra grape extract (samples CG.F20 and CG.F30) exhibited a DPPH scavenger activity greater than 77% at the higher concentrations (Figure 5). However, for the CG.F30 sample, the potential antioxidant properties was observed to appear after 5 min of reaction, while for the CG.F20 sample, the effect was obvious at 30 min of reaction. A similar behavior was visible at high dilutions (500 and 750 µL/mL corresponding to the concentration between 979 and 1635.5 µg/mL) (Figure 5) for CG.F20 and CG.F30 samples, but overall for the CG.F20 sample, although the antioxidant effect was installed more slowly, the increase in the antioxidant activity was not acute (sharp) between the different dilutions, but appeared in a progressive manner. For the CG.F10 sample with 10 *v*/*v*% Feteasca Neagra extract incorporated, an inhibitory activity of DPPH radicals below 40% was recorded at the tested dilutions and only after 1 h of reaction. 

In the case of composite cryogels with Merlot grape extract, the samples showed a weaker antioxidant activity (inhibition of free radicals DPPH below 50%) (CG.M20, CG.M30, CG.M40) or even absence (CG.M10) (Figure 6). The antioxidant effects were installed more slowly, and only after 1 h of reaction. The CG.M30 and CG.M40 samples containing the highest concentrations of incorporated extract (30 *v*/*v*% and respectively 40 *v*/*v*%) exhibited an activity of about 28% and respectively, 38% at higher dilutions (Figure 6).

In general, polyphenols possess strong antioxidant properties, and their antioxidant activity depends on the number and position of -OH groups on the aromatic ring [32]. Notably, the inclusion of tannic acid [11] or flavones [32], significantly enhanced the DPPH radical scavenging activity of the composite films.

In conclusion, the degree of antioxidant capacity of XG/PVA/grape pomace extract composite cryogels was proportional to the total phenolic content of grape variety and to their amount added into the matrices. The results obtained in this study are in agreement with other research studies reported for chitosan-based films embedding *Vitis vinifera* grape pomace extracts [33], proanthocyanidins [34], flavones (chrysin, apigenin and luteolin) and D-α-tocopheryl polyethylene glycol 1000 succinate [32], and tannic acid [11].

#### 2.2.2. Antibacterial Activity

Due to the increased resistance of pathogens to common antibiotics and disinfectants, current research deals with the development of new materials with antimicrobial properties [35]. Hydrogel-based materials containing various antioxidants may be considered suitable alternatives. In this regard, the feature of XG-based and PVA-based cryogels to inhibit the growth of two Gram-negative bacteria, namely *Escherichia coli* and *Salmonella typhymurium*, and a Gram-positive bacterium, *Listeria monocytogenes*, was investigated. Table 3 and Figure 7, Figure 8 and Figure 9 show the percentage of the bacterial inhibition and the optical images of bacterial colonies developed in the absence (control) and presence of XG/PVA-based composite films containing extracts from Feteasca Neagra (CG.F10–CG.F30) and Merlot (CG.M10–CG.M40) grape pomace.

Antibacterial tests showed that samples containing the hydro-alcoholic extract of Feteasca Neagra (CG.F10–CG.F30) and Merlot (CG.M10–CG.M40) pomace exhibited more pronounced antibacterial activity compared to the polymeric matrix based on XG and PVA (except samples containing Feteasca Neagra in the case of *Listeria monocytogenes*) (Table 3). The hydro-alcoholic extract of the Merlot pomace was more effective as an inhibitor for the growth of bacterial colonies, without a clearly defined dependence on concentration (Figure 7, Figure 8 and Figure 9). This behavior could be associated with the pronounced hydrophobic character observed for these samples (low water swelling and high contact angle values, Table 1 and Figure 2).

By adding the hydro-alcoholic extract of Feteasca Neagra to the XG/PVA-based cryogels, bacterial inhibition was more pronounced in Gram-negative bacteria (*Salmonella typhymurium* and *Escherichia coli*). Moreover, the percentage of inhibition in the case of the bacterium *Listeria monocytogenes* of the XG/PVA cryogel decreased by embedding the extract from the Feteasca Neagra pomace; a slight increase in the percentage of inhibition was observed by raising the concentration of the added extract (Table 3). This may be due to chemical interactions that occur between XG/PVA and the compounds present in the Feteasca Neagra extract, which led to a blocking effect of the chemical groups capable of disrupting/inhibiting the growth of the bacterium.

All XG/PVA-based cryogels containing hydro-alcoholic extract from Merlot pomace (CG.M10–CG.M40) had higher antibacterial activity (for all three bacteria tested) compared to the XG/PVA cryogel without extract (sample CG, Table 3). In this case, also, a greater sensitivity of Gram-negative bacteria to these newly obtained materials was observed. In addition, the percentage of the bacterial inhibition of XG/PVA-based composite films containing hydro-alcoholic extract from Feteasca Neagra and Merlot grape pomace achieved onto *Escherichia coli* and *Listeria monocytogenes* (Table 3) was superior to those already reported for other polysaccharide-based films incorporating quercetin or tannic acid [36] and were comparable only with those embedding pyocyanin [37]. Moreover, as far as we know, this is the first study which reports XG/PVA/Merlot pomace-based cryogels with an antimicrobial activity of 99% against *Salmonella typhymurium*.

The antibacterial mechanism of XG is based on the electrostatic interactions between the functional groups of glucuronic acid and pyruvic acid and the positively charged residues on the surface of microorganisms, leading thus to the leakage of intracellular components [27]. By the addition of Feteasca Neagra or Merlot grape extracts, the antibacterial activity of the composite cryogels is significantly enhanced. This behavior could be explained by their ability to capture iron, which is necessary in the growth of bacteria, or the binding of these compounds to vital proteins (such as microbial enzymes) by hydrogen bonds [38].

The highly hydrophobic character of XG/PVA composite films containing Merlot grape pomace extract and their remarkable antimicrobial activity indicates a great potential of these materials for food packaging applications. 

### 2.3. Effect of Variety of Grape Pomace Extracts and Their Amount on the Texture of Composites 

The properties of composite cryogels are also tightly linked to their morphology and texture. In this context, the internal microstructure of composites was analyzed by SEM. In Figure 10 and Figure 11, the morphology of pure PVA and XG cryogels is presented in comparison to that of XG/PVA composite cryogel, without and with Feteasca Neagra (samples CG.F10–CG.F30) or Merlot (samples CG.M10–CG.M40) pomace extracts. 

All cryogels showed a heterogeneous morphology with interconnected pores (Figure 10 and Figure 11), which is a result of the lyophilization process applied after sample preparation. The addition of Feteasca Neagra or Merlot pomace extracts generates composite cryogels with a rough microstructure and well-defined pores, depending on the amount of Feteasca Neagra or Merlot pomace extract embedded within the XG/PVA 3D networks. A uniform distribution of the pores was obtained when 20 *v*/*v*% of Feateasca Neagra pomace extract (Figure 10E) or 30 *v*/*v*% of Merlot pomace extract (Figure 11C) was encapsulated within the XG/PVA matrix. A similar phenomenon was noticed when flavones [32] or baicalein [39] were incorporated into polysaccharide-based composite films.

In addition, the pore walls of the composite cryogels (Figure 10C) are less compact and thus more accessible for diffusion and strong interaction with low molecular weight bioactive species, such as polyphenols found within our grape pomace extract samples.

The average pore diameters of all samples (Table 4) were determined by Image J 1.48v analyzing software, measuring at least 20 pores (voids) on each of three independent SEM micrographs. 

As Table 4 shows, the average pore (voids) sizes of the composite cryogel (sample CG) without pomace extracts are around 117 ± 22 µm, slightly lower than that of pure XG or PVA cryogels. The addition of 10 *v*/*v*% Feteasca Neagra extract led to changes only on the morphology of the pore walls, as the pore sizes diameters did not change too much. A uniform distribution of smaller pores with an average diameter 72 ± 11 µm was observed when 20 *v*/*v*% Feteasca Neagra extract was added to the XG/PVA cryogels. A further increase of the Feteasca Neagra extract within the XG/PVA matrix (sample CG.F30) significantly affected the pore walls morphology as well as their sizes, which increased to 137 ± 21 µm. In the case of composite cryogels entrapping Merlot pomace extract, the pore diameters decreased with the increase of the extract content, i.e., from 103 ± 17 µm (sample CG.M10) to 60 ± 8 µm (sample CG.M30). However, an additional increase of Merlot extract content to 40 *v*/*v*% induced modifications of the pore walls morphology and of the pore diameters, which rose to 90 ± 12 µm. Thus, porous composite materials with various pore sizes were facile designed by ranging the concentration of the pomace extract and the grape variety within the 3D matrices. These results allow us to envisage their potential future application as scaffolds in tissue engineering. 

## 3. Conclusions

Bioactive composite cryogels were successfully prepared by incorporating two varieties of grape pomace extracts (Feteasca Neagra and Merlot) into ice-templated XG/PVA networks. The hydrophobic/hydrophilic properties, the texture and pore sizes, the radical scavenging activity and the antibacterial properties of XG/PVA/grape pomace extract were greatly influenced by the type of extract. The composite samples containing 30 *v*/*v*% Feateasca Neagra grape extract and 40 *v*/*v*% Merlot grape extract presented the lowest water swelling ratios (15.8 g/g and, respectively, 11.19 g/g) and the highest water contact angle values (89.08° and, respectively, 90.95°). The significant improvement of surface hydrophobicity for the cryogel films was attributed to the incorporation of polyphenols, i.e., *trans*-resveratrol, which forms hydrogen bonding with the XG/PVA matrix. Highly hydrophobic films are preferable in food packaging applications because they preserve the quality and extend the shelf-life of products. 

The films with Feteasca Neagra type grape pomace extract exhibited an extraordinary antioxidant activity, this behavior being associated with the total phenolic content, which is higher in the case of Feteasca Neagra grape variety. However, the XG/PVA-based cryogels containing hydro-alcoholic extracts from Merlot pomace had superior antibacterial activity toward Gram-negative bacteria (*Escherichia coli* and *Salmonella typhymurium*) and a Gram-positive bacterium (*Listeria monocytogenes*) compared to the XG/PVA cryogel without the extract. To date, as far as we know, the remarkable antimicrobial activity against *Salmonella typhymurium* exhibited by XG/PVA/Merlot pomace-based cryogels is for the first time reported. This result was assigned to the high hydrophobicity exhibited by these composites (low values for water absorbency and high values for water contact angle). 

Our study pointed out that eco-friendly porous composite cryogels with controlled/desired features could be developed by an adequate selection of the grape variety and its concentration within the polymeric matrix. Moreover, by ranging the concentration of pomace extract and grape variety within 3D matrices, composite cryogels with a rough microstructure and well-defined pores of various sizes were facile designed, using lyophilization as a drying method. Composite cryogels with a uniform distribution of the pores were achieved when 20 *v*/*v*% of Feateasca Neagra pomace extract or 30 *v*/*v*% of Merlot pomace extract was entrapped within the XG/PVA matrix. These findings allow us to extend the application range of these composites and to recommend them as useful candidates in tissue engineering.

## 4. Materials and Methods

### 4.1. Materials

Xanthan gum (XG), and poly(vinyl alcohol) (PVA) were purchased from Sigma-Aldrich Chemie GmbH (Schnelldorf, Germany) and both were used as received. The average molecular weight of XG (M_v_ = 1.98 × 10^6^ g mol^−1^) was determined by viscometry, according to a previously reported procedure [40]. It was established by the Sloneker and Orentas method [41] that 1 in 3 repeat units were pyruvate substituted, while back-titration indicated 2 out of 3 saccharide repeat units were acetylated. The number-average molar mass and polydispersity index of PVA (M_n_ = 96,619 g mol^−1^; PDI = 1.17) were evaluated by gel permeation chromatography. The grape pomace samples (Feteasca Neagra or Merlot type) were harvested in the autumn of 2019 at full maturity from Iasi vineyard and were provided by “Ion Ionescu de la Brad” University of Agricultural Sciences and Veterinary Medicine. The grapes were crushed and peeled following the maceration process (5 days at 10 °C), at the end of which the pressing operation followed. The resulting pomace from the pressing operation was dried and stored in the freezer until the moment of analysis. The grape pomace extracts were prepared, according to a previously reported extraction method in ethanol [31,42]. Briefly, the extraction was performed, using a mixture of 50% water and 50% ethanol. The extracts were prepared by mixing 2 g of solid sample (batch) with 5 mL of H_2_O/EtOH mixture, followed by stirring for 24 h at room temperature, and then centrifuged at 4500 rpm for 20 min at 4 °C. The supernatants were recovered and used for further experiments. The chemical composition of the grape pomace samples was already investigated by high performance liquid chromatography along with their antioxidant and antimicrobial properties [31,42]. The total phenolic content of the Feteasca Neagra and Merlot extracts were evaluated by the Folin–Ciocalteu method, as 1.452 and, respectively, 2.42 mg gallic acid/mL. 

### 4.2. Methods

#### 4.2.1. Preparation of Ice-Templated 3D Cryogels

The 3D composite cryogels consisting of XG, PVA, and Feteasca Neagra or Merlot grape pomace extracts were obtained by the freeze/thawing approach, using the optimum reaction conditions identified previously [27]. The weight ratio between XG and PVA (1:1) and the freeze/thawing conditions (seven freeze/thawing cycles, −20 °C as freezing temperature for 24 h, 22 °C as thawing for 12 h) were maintained constant during the preparation of all samples. The parameters varied upon cryogel fabrication were the grape variety (Feteasca Neagra or Merlot) and their content within the XG/PVA networks (10 to 40 *v*/*v*%). All samples were dried by lyophilization, using a LABCONCO FreeZone device (Kansas City, MO, Fort Scott, KS, USA).

#### 4.2.2. Gel Permeation Chromatography (GPC)

GPC measurements (number-average molar mass (M_n_)), weight-average molar mass (M_w_), and molar mass distribution (M_w_/M_n_) (PDI) for PVA were performed in water, using two Viscotek CLM 3021 A 6000M columns and a RI Detector 2300, Knauer. Polyethylene oxide standards were used for calibration.

#### 4.2.3. Water Swelling Analysis

The water swelling behavior of XG-based cryogels was gravimetrically evaluated by immersing a certain amount of dried cryogels (0.01 g) in 20 mL aqueous solutions. The swelling ratio (*SR*, g/g) was calculated as follows:(1)$$SR=\frac{W_{t}}{W_{d}}$$where *W_t_* is the weight of swollen film at time *t*, and *W_d_* is the weight of the freeze-dried sample. 

#### 4.2.4. Water Contact Angle Measurements

The static contact angle (θ) measurements were performed by the drop method [43] for all films. Briefly, 1 μL of MilliQ water was placed on the surface of each composite film, using a CAM-200 instrument. The θ values were obtained by fitting the Young–Laplace equation onto the drop profile of each film, and they were calculated as an average of five consecutive measurements.

#### 4.2.5. Morphological and Texture Studies

The cross-sectional microstructure of the film was observed by a Quanta 200-FEI-type environmental scanning electron microscope (ESEM, FEI Company, Hillsboro, OR, USA) at 20 kV in low vacuum mode, whilst the average pore diameters of all samples were determined by Image J 1.48v analyzing software (National Institutes of Health and the Laboratory for Optical and Computational Instrumentation, University of Wisconsin, Madison, WI, USA) [44]. To evaluate the diameter of pores, we measured at least 20 pores (voids) on each of three independent SEM micrographs taken for every cryogel sample. 

#### 4.2.6. Elemental Surface Composition

The energy dispersive X-ray (EDX) detector was used to map the elements and to estimate their ratio and distribution within grape pomace extracts-loaded XG/PVA matrices.

#### 4.2.7. Thermogravimetric (TGA) and Derivative Thermogravimetric (DTG) Analysis

The TGA experiments were performed on a STA 449F1 Jupiter (NETZSCH-Gerätebau GmbH, Selb, Germany) analyzer. The mass loss of the XG/PVA sample and of the grape pomace extract-loaded films was measured between 25 °C and 600 °C under N_2_ gas, at a flow rate of 50 mL/min.

#### 4.2.8. DPPH Radical Scavenging Activity

The in vitro antioxidant activity of XG-based cryogel composites was assayed, according to a previously reported procedure [27]. Typically, 50 mg of each film sample was added to 3 mL of 50% ethanol, and after vortexing, 1 mL of ultrapure water was added and vortexed again. After 24 h, 1.5–3.5 mL of ultrapure water was added to the sample, and then the obtained dispersions were subjected to ultracentrifugation after which the supernatant was used in the DPPH test. From each sample (supernatant), dilutions of 250, 500 and 750 µL/mL in ultrapure water were made. An additional dilution of 125 µL/mL was performed for the sample CG.F20. To 500 µL of each dilution was added 2000 µL of DPPH methanolic solution (0.06 mM). The mixture was well blended and left to stand in the dark. Depending on the sample analyzed, after 5 min, 30 min or one hour, the absorbance was measured at λ = 517 nm. The free radical inhibiting activity (%) was calculated as follows:(2)$$DPPH\;scavengigng\;activity,\;\%=\;\frac{Ac-As}{Ac}\times 100$$
where *A_C_* is the absorbance of control sample (2000 µL DPPH methanolic solution and 500 µL methanol), and *A_S_* is the absorbance of tested sample.

#### 4.2.9. Testing the Antimicrobial Activity against Gram-Positive and Gram-Negative Microorganisms

The antibacterial activity of the control and of the grape pomace extracts-loaded cryogels was tested against two Gram-negative bacteria, (*Escherichia coli* and *Salmonella typhymurium*) and one Gram-positive bacterium, *Listeria monocytogenes*, following the previously published methodology [27]. Briefly, suspensions from ATCC strains were prepared in peptonate saline (physiological serum) with a turbidity of 1° McFarland. Then, several dilutions were prepared with a concentration of 1500 colony forming units (CFU)/mL. Afterward, the surfaces of the tested samples and of the wrapping paper as a control were contaminated with 100 µL of ATCC strain, and kept in contact for 24 h. Finally, the inoculum was extracted, soaked in peptone saline, and seeded on the surface of the specific medium (XLD, VRBG, ALOA). The plates were incubated in a thermostat at 37 ± 1 °C for 24 h. The colonies were counted, and the values were compared with the control (SR ISO 7218—General Directive for microbiological examinations; SR ISO 11133—Guide for the preparation and production of culture media) [43].

## Figures and Tables

**Figure 1 gels-07-00076-f001:**
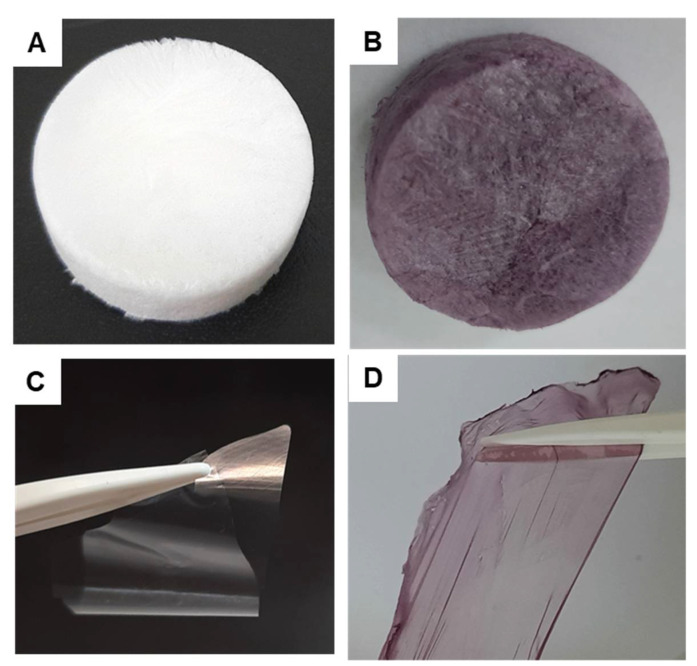
Optical images of the XG/PVA (**A**,**C**) and CG.F10 (**B**,**D**) composite cryogels dried by lyophilization (**A**,**B**) or by solvent casting (**C**,**D**).

**Figure 2 gels-07-00076-f002:**
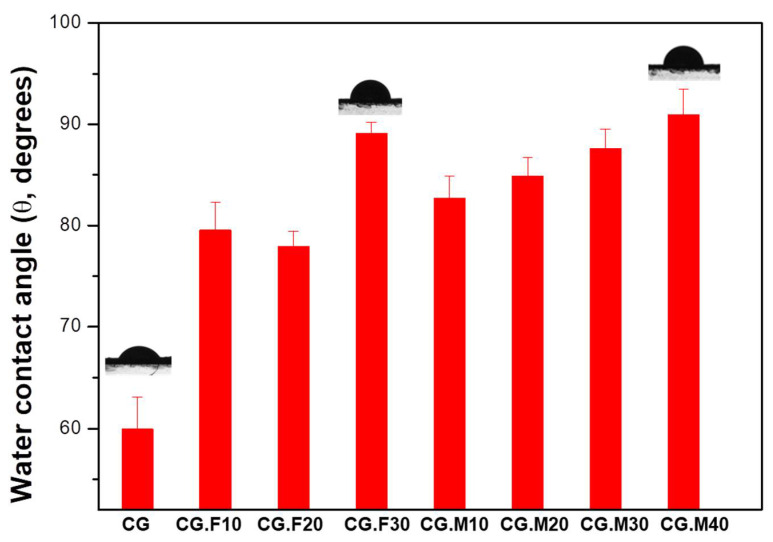
Water contact angle of the 3D cryogel matrix without grape pomace extract (sample CG) and of the composite cryogels containing different type and amount of grape pomace extract (samples CG.F10–CG.F30 for Feteasca Neagra grape type; and samples CG.M10–CG.M40 for Merlot grape type).

**Figure 3 gels-07-00076-f003:**
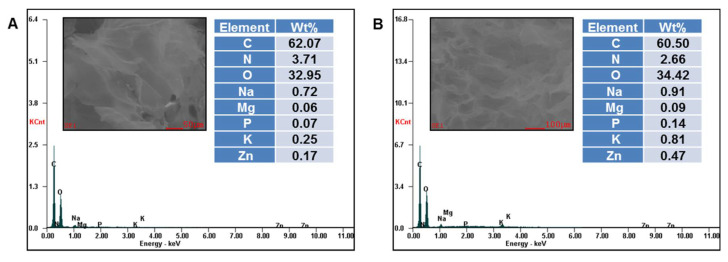
EDX profiles of CG.F10 (**A**); CG.F20 (**B**) and CG.F30 (**C**) composite cryogels.

**Figure 4 gels-07-00076-f004:**
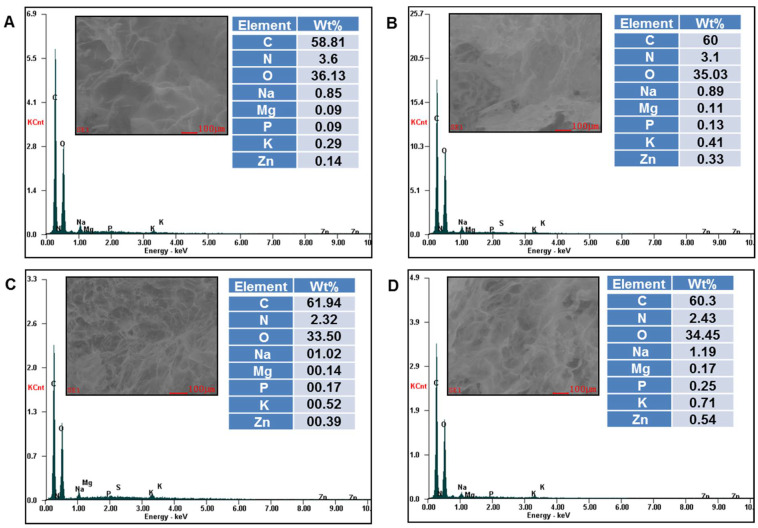
EDX profiles of CG.M10 (**A**); CG.M20 (**B**); CG.M30 (**C**) and CG.M40 (**D**) composite cryogels.

**Figure 5 gels-07-00076-f005:**
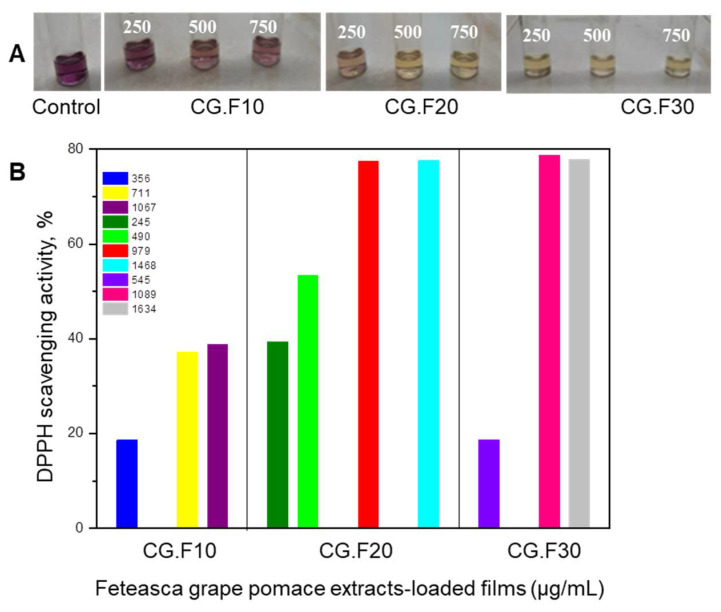
(**A**) Optical images of Control (XG/PVA composite without Feteasca Neagra grape pomace extracts), CG.F10, CG.F20, and CG.F30 samples after applying DPPH test; (**B**) DPPH radical scavenging activity of Feteasca Neagra grape extract-loaded films performed after 1 h (CG.F10), 30 min (CG.F20), and 5 min (CG.F30) of incubation in dark.

**Figure 6 gels-07-00076-f006:**
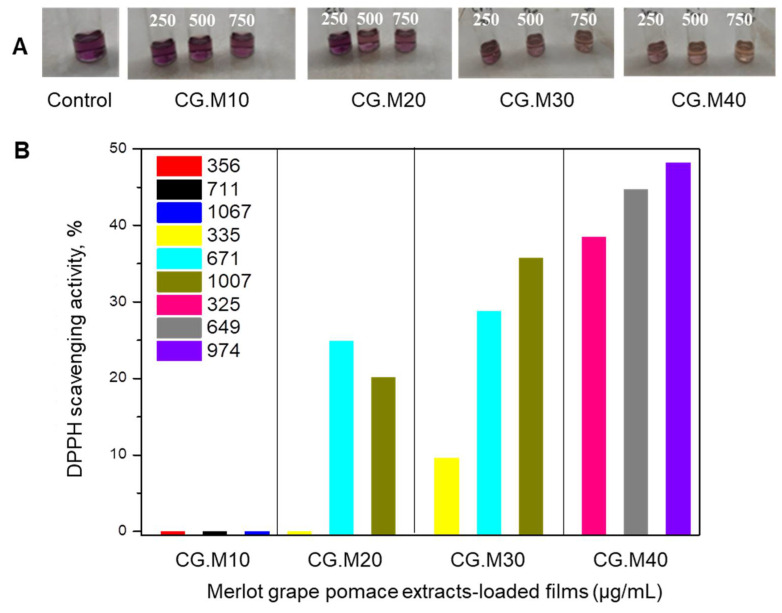
(**A**) Optical images of control (XG/PVA composite without Merlot grape pomace extracts), CG.M10, CG.M20, CG.M30, and CG.M40 samples after applying DPPH test; (**B**) DPPH radical scavenging activity of Merlot grape extract-loaded films performed after 1 h (CG.M10), 30 min (CG.M20), and 5 min (CG.M30) of incubation in dark.

**Figure 7 gels-07-00076-f007:**
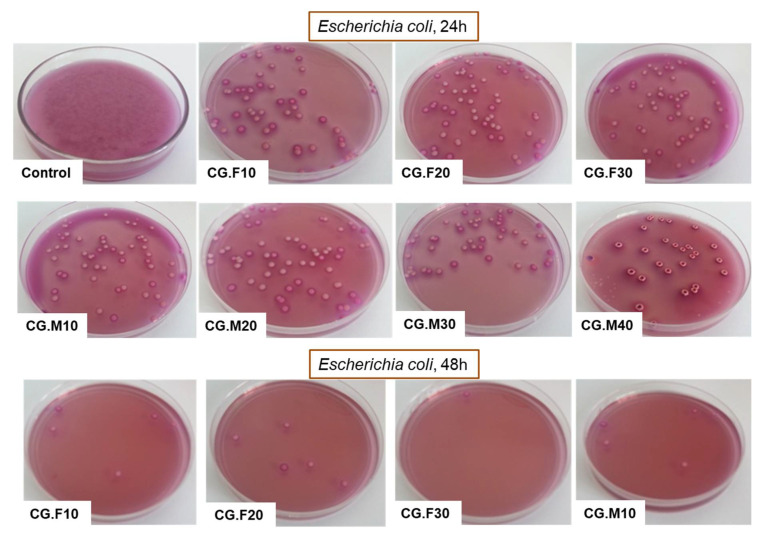
Optical images of *Escherichia coli* bacterial colonies developed in the absence (control) and presence of XG/PVA-based composite films containing hydro-alcoholic extracts from Feteasca Neagra (CG.F10-CG.F30) and Merlot (CG.M10-CG.M40) pomace.

**Figure 8 gels-07-00076-f008:**
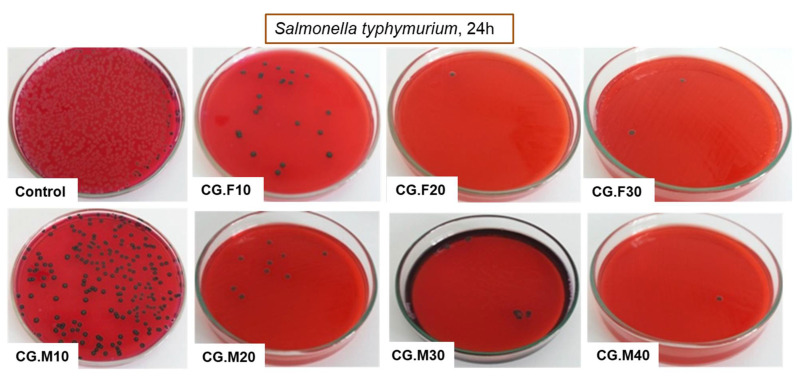
Optical images of *Salmonella typhymurium* bacterial colonies developed in the absence (control) and presence of XG/PVA-based composite films containing hydro-alcoholic extracts from Feteasca Neagra (CG.F10–CG.F.30) and Merlot (CG.M10–CG.M40) pomace.

**Figure 9 gels-07-00076-f009:**
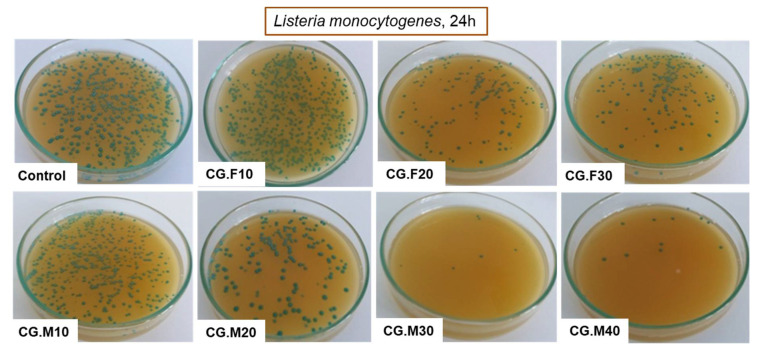
Optical images of *Listeria monocytogenes* bacterial colonies developed in the absence (control) and presence of XG/PVA-based composite films containing hydro-alcoholic extracts from Feteasca Neagra (CG.F10–CG.F30) and Merlot (CG.M10–CG.M40) pomace.

**Figure 10 gels-07-00076-f010:**
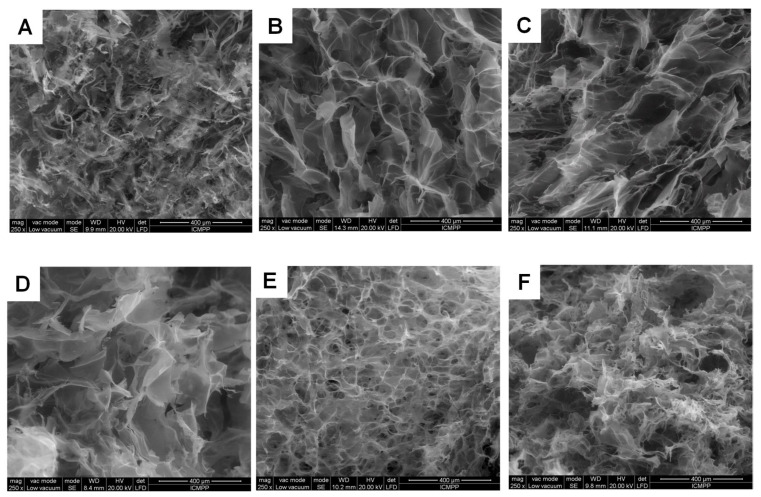
SEM micrographs of PVA (**A**); XG (**B**); XG/PVA (**C**); CG.F10 (**D**); CG.F20 (**E**) and CG.F30 (**F**) cryogels.

**Figure 11 gels-07-00076-f011:**
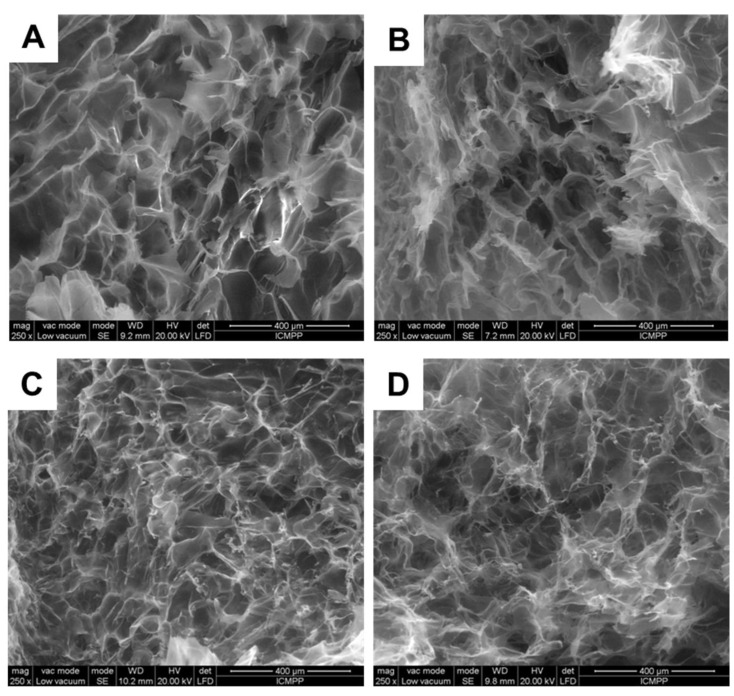
SEM micrographs of CG.M10 (**A**); CG.M20 (**B**); CG.M30 (**C**) and CG.M40 (**D**) cryogels.

**Table 1 gels-07-00076-t001:** The sample codes, composition, and some characteristics of bioactive composite cryogels.

Samples ^a^	XG, wt.%	PVA, wt.%	Amount of Grape Pomace Extracts, *v*/*v*%	SR ^b^, g/g
CG	50	50	0	32.14 ± 1.22
CG.F10	50	50	10	21.92 ± 2.60
CG.F20	50	50	20	20.92 ± 1.71
CG.F30	50	50	30	15.80 ± 2.05
CG.M10	50	50	10	20.00 ± 0.94
CG.M20	50	50	20	16.55 ± 1.14
CG.M30	50	50	30	13.25 ± 0.88
CG.M40	50	50	40	11.19 ± 1.47

^a^ The code sample of composite films includes the following: “CG” from composite gel; “F” or “M” from the variety of the grape pomace extract, Feteasca Neagra or Merlot. The final number from the sample codes represents the amount of extract entrapped within the composite matrices. All samples were prepared upon seven freeze-thawing cycles. ^b^ SR represents the swelling ratio and was evaluated by Equation (1) (see Section 4, Materials and Methods).

**Table 2 gels-07-00076-t002:** Data extracted from the TGA and DTG curves of the composite cryogels without (sample CG) and with Feteasca Neagra extract incorporated (sample CG.F10).

Sample	Stage	*T*_onset_ (°C)	*T*_max_ (°C)	*T*_endset_ (°C)	*W*_m_ (%)	*W*_rez_ (%)
CG	I	38	79	112	19.96	
	II	237	301	330	56.56	
	III	377	434	479	14.17	18.3
CG.F10	I	42	64	101	6.53	
	II	171	203	235	24.75	
	III	262	296	345	24.75	
	IV	345	440	481	13.54	28.9

*T*_onset_—onset temperature of thermal degradation; *T*_max_—temperature that corresponds to the maximum rate of thermal decomposition for each stage evaluated from the peaks of the DTG curves; *T*_endset_—endset temperature of thermal degradation; *W*_m_—mass loss rate corresponding to each thermal degradation stage; *W*_rez_—percentage of residue remained at the end of thermal degradation (700 °C).

**Table 3 gels-07-00076-t003:** The percentage of the bacterial inhibition of XG/PVA-based composite films containing hydro-alcoholic extract from Feteasca Neagra (EF) and Merlot (EM) grape pomace.

Sample	Inhibition (%) *Escherichia coli*		Inhibition (%) *Salmonella typhymurium*	Inhibition (%) *Listeria monocytogenes*
	24 h	48 h	24 h	24 h
Control	0	0	0	0
CG	35	-	26	77
CG.F10	66	18	86	18
CG.F20	68	22	94	22
CG.F30	76	41	95	41
CG.M10	75	67	43	67
CG.M20	74	71	99	71
CG.M30	71	96	99	96
CG.M40	86	93	99	93

**Table 4 gels-07-00076-t004:** The average pore sizes of XG-based composite cryogels.

Samples	PVA	XG	CG	CG.F10	CG.F20	CG.F30	CG.M10	CG.M20	CG.M30	CG.M40
**Pore sizes, µm**	138 ± 25	125 ± 22	117 ± 22	114 ± 12	72 ± 11	137 ± 21	103 ± 17	86 ± 6	60 ± 8	90 ± 12

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
