# Peer review of "A Comparative Analysis on the Effect of Variety of Grape Pomace Extracts on the Ice-Templated 3D Cryogel Features"

_gels, 2021, doi:10.3390/gels7030076_

Round 1

Reviewer 1 Report

The paper deals with the prepared hydrogels-xerogels. It is well written, the results are well discussed. The only issue is connected to the novelty of the proposed approach and received result in comparison to the literature. It is recommended to strengthen the abstract, introduction and conclusions regarding this concern.

Author Response

Thank you for your letter and for the reviewers’ comments concerning our manuscript entitled “A Comparative Analysis on the Effect of Variety of Grape Pomace Extracts on the Ice-Templated 3D Cryogel Features”.

The reviewer’s comments were very helpful for improving our paper. We have studied all comments carefully and made the requested corrections.

The main changes were highlighted in red in the manuscript and the reviewers’ comments were addressed as presented below:

Reviewer #1

The paper deals with the prepared hydrogels-xerogels. It is well written, the results are well discussed. The only issue is connected to the novelty of the proposed approach and received result in comparison to the literature. It is recommended to strengthen the abstract, introduction and conclusions regarding this concern.

Response: The authors thank to the reviewer for this kind suggestion. The abstract, introduction, and conclusions have been upgraded to highlight the novelty and the most significant finding of this work. In addition, the discussion on antioxidant and antimicrobial properties was further improved and sustained by citing new references:

  1. Andreia S. Ferreira, Cláudia Nunes, Alichandra Castro, Paula Ferreira, Manuel A. Coimbra Influence of grape pomace extract incorporation on chitosan films properties. Carbohydrate Polymers 113 (2014) 490–499.
  2. Mojca Bozic, Selestina Gorgieva, Vanja Kokol Homogeneous and heterogeneous methods for laccase-mediated functionalization of chitosan by tannic acid and quercetin. Carbohydrate Polymers 89 (2012) 854– 864.
  3. Fengyu Bi, Xin Zhang, Ruyu Bai, Yunpeng Liu, Jing Liu, Jun Liu Preparation and characterization of antioxidant and antimicrobial packaging films based on chitosan and proanthocyanidins. International Journal of Biological Macromolecules 134 (2019) 11–19.
  4. Sunirmal Sheet, Mohanraj Vinothkannan, Saravanakumar Balasubramaniam, Sivakumar Allur Subramaniyan, Satabdi Acharya, and Yang Soo Lee. Highly flexible electrospun hybrid (polyurethane/dextran/pyocyanin) membrane for antibacterial activity via generation of oxidative stress. ACS Omega 2018, 3, 14551−14561.

Reviewer 2 Report

The manuscript details an interesting use of grape pomace, namely as an additive in biocompatible food packaging. The manuscript is mainly well written and can be interesting for the readers of Gels, therefore I suggest its acceptance, however after a major revision.

My comments to the authors are as follows:

Major issues:

The conclusion part is too short and should be extended.

For practical applications the look of the packaging material is also a key aspect. Please include pictures of the composites at least in the Supporting Info.

Minor issues

  • Line 12, “increase” instead of increased
  • Line 28, “Currently, the wine from the winemaking” I think You meant pomace here instead of wine. The same goes for line 34.
  • Line 105, The molecular weight varies between 2*10^6 and 20*10^6 Da and not its distribution. It should be clarified which type of molecular weight (number or weight average) is given and the polydispersity of XG and PVA should be added. Furthermore, Mw also has the unit of Da.
  • Line 149, the units of contact angle should be given.
  • Line 180, What is DPPH?

Author Response

Thank you for your letter and for the reviewers’ comments concerning our manuscript entitled “A Comparative Analysis on the Effect of Variety of Grape Pomace Extracts on the Ice-Templated 3D Cryogel Features”.

The reviewer’s comments were very helpful for improving our paper. We have studied all comments carefully and made the requested corrections.

The main changes were highlighted in red in the manuscript and the reviewers’ comments were addressed as presented below:

Reviewer #2

The manuscript details an interesting use of grape pomace, namely as an additive in biocompatible food packaging. The manuscript is mainly well written and can be interesting for the readers of Gels, therefore I suggest its acceptance, however after a major revision.

My comments to the authors are as follows:

Major issues:

The conclusion part is too short and should be extended.

Response: The authors appreciate the reviewer’s suggestion. In this regard, the conclusion part has been extended.

For practical applications the look of the packaging material is also a key aspect. Please include pictures of the composites at least in the Supporting Info.

Response: The authors thank to the reviewer for this kind suggestion. Therefore, optical images of the XG/PVA (images A, C) and CG.F10 (images B,D) composite cryogels dried by lyophilization (images A and B) or by solvent casting (images C and D) have been included as Figure 1 in revised manuscript.

Minor issues

  • Line 12, “increase” instead of increased

Response: The authors agreed with the kind recommendation of the reviewer. Consequently, the word “increased” has been replaced with “increase”.

  • Line 28, “Currently, the wine from the winemaking” I think you meant pomace here instead of wine. The same goes for line 34.

Response: The authors agreed with the reviewer comment. Therefore, the word “wine” has been replaced with “pomace”.

  • Line 105, The molecular weight varies between 2*10^6 and 20*10^6 Da and not its distribution. It should be clarified which type of molecular weight (number or weight average) is given and the polydispersity of XG and PVA should be added. Furthermore, Mw also has the unit of Da.

Response: The authors agreed with the kind recommendation of the reviewer. In this regard, the molecular weight of XG and PVA and the methods used for their determination were included, in the revised manuscript, in the “Materials and Methods” section.  

  • Line 149, the units of contact angle should be given.

Response: The contact angle unit has been added in the revised manuscript.

  • Line 180, What is DPPH?

Response: DPPH is 2,2-diphenyl-1-picrylhydrazyl and this information was also included in the revised manuscript.

Round 2

Reviewer 2 Report

The authors made all the corrections requested, therefore I suggest the acceptance of the manuscript.